# Let Him Who Is without Sin Cast the First Stone: Religious Struggle among Persons Convicted of Sexually Offending

**Theresa M. Robertson [1,*], Gina M. Magyar-Russell [2] and Ralph L. Piedmont [3]**

[1]  Independent Scholar, Mill Centre in Baltimore, MD 21211, USA
[2]  Department of Psychology, Loyola University Maryland, Baltimore, MD 21210, USA; gmmagyarrussell@loyola.edu
[3]  Center for Professional Studies, Timonium, MD 21093-9998, USA; ralphpiedmont01@gmail.com
*  Correspondence: tmrobertson@loyola.edu

**Abstract:** Religiousness and spirituality have been identified as important factors in promoting desistance from sexual offending and as helpful coping resources with negative psychological consequences related to public registration. However, the potential mental health benefits, and detriments, of religiousness and spirituality for persons convicted of sexually offending have not been widely examined. Given the moral implications of their behavior and stigmatization by society, including from religious and spiritual communities, this study aimed to examine levels of religious struggle and their associations with symptoms of mental health among 30 men on the Maryland Sex Offense Registry. Relative to the normative sample, the mean level of spiritual transcendence, constructive perceptions of spirituality that develop within social, cultural, and educational contexts, was significantly lower within this sample. Conversely, religious struggle mean scores indicated that the men in this sample experienced significantly greater difficulties relative to God and their faith community. Greater levels of religious struggle were significantly related to higher neuroticism, greater self-reported shame, depression, anxiety, and hopelessness, as well as lower levels of self-compassion. Based on these preliminary findings, religious struggles may adversely influence the mental health of persons convicted of sexually offending. More research is needed to gain a better understanding of the associations between religiousness, spirituality, and mental health in this population. Future directions for research and clinical implications for mental health providers, including spiritually informed treatment approaches with persons convicted of sexually offending, are discussed.

**Keywords:** religious and spiritual struggles; sex offenders; sexual offending; faith communities; compassion; self-compassion

## 1. Introduction

> "The one who bears the sore of leprosy shall keep his garments rent and his head bare, and shall muffle his beard; he shall cry out, 'Unclean, unclean!' As long as the sore is on him, he shall declare himself unclean, since he is in fact unclean. He shall dwell apart, making his abode outside the camp." (Leviticus 13:45–46, New American Bible)

There are many similarities between the leprosy laws and current sex offense laws. Sex offense registries are the modern-day equivalent of the Old Testament law, requiring persons with the disease of leprosy to warn others of the threat posed by their presence. Residency restrictions force sex offenders to make their "abode outside the camp," ostensibly to protect the public. Leprosy laws were based on a number of false assumptions, as are sex offense laws. Both leprosy laws and sex offense

laws have resulted in stigmatization resulting in difficult experiences and emotions such as isolation, shame, and hopelessness.

An important difference between the ancient leprosy laws and current sex offense laws is that people with leprosy had a physiological disease. They did not make a conscious choice to contract the disease, and the social stigma that they were subjected to was unfair, as their disease was not their fault. On the other hand, it could be argued that those who have sexually offended have acted upon decisions to engage in behaviors that have been deemed unacceptable and illegal, and therefore the stigmatization they experience is their fault. However, sexual offending is not as straightforward as contracting a bacterial infection.

The etiology of sexual offending is not well understood. Treatment options are limited, and environmental, physiological, and psychological factors are not considered when labeling individuals as "sex offenders." Although it is assumed that all those who have sexually offended are blameworthy, there are mitigating factors that render this assumption inaccurate, just as the ostracism and treatment of persons with leprosy at the time of Jesus was based on inaccurate assumptions at that time and place in history.

Misconceptions about the causes and risks of sexual offending abound and have led to misinformed assignment of blame (Moen 2015), public panic (Berryessa 2014), and the adoption of laws, policies, and public perceptions that are unnecessarily punitive and ineffective in reducing potential harm to others (Tofte 2007). Moen (2015) argued that even when a wrong sexual act has been committed there are factors that might mitigate blame. Specifically, Moen pointed to the appeal to ignorance and the appeal to moral luck. The appeal to ignorance refers to the offender who is unaware of the potential for significant risk of harm to another resulting from his or her act. It should be noted that the appeal to ignorance can also be a form of denial. This is an important distinction in assessing blameworthiness, as well as a possible consideration in the process of negotiating the stigma associated with sexually offending that is often present in the initial phases of desistence (Hulley 2016). Within the context of blameworthiness an example of the appeal to ignorance could be applied to children engaging in explorative sexual acts or teenagers engaging in consensual sex.

Moen (2015) appeal to moral luck can be viewed as having the misfortune of being born with a sexual preference for minor children. While the precise etiologies of hebophilia and pedophilia remain uncertain, they are believed to result from an unknown confluence of psychological, social, and neurobiological influences (Berryessa 2014; Labelle et al. 2012). Phenomenologically, people with a diagnosis of pedophilia report they do not choose to be sexually attracted to children, rather they discovered a minor attraction in their teen years (Moen 2015). There is no evidence that sexual preference for minors is a choice, and Alanko, Salo, Mokros, and Santtila (Alanko et al. 2013) found evidence among a sample of nearly 4000 Finnish twins and male siblings that suggest genetic factors may contribute to minor attraction. This argument is not meant to imply that adult–child sex is not wrong or harmful, or that it should not be prosecuted, but rather to provide a framework for reassessing individual blameworthiness. It should be noted that the common notion that anyone who is attracted to minors either has offended or will offend if they have not already done so is false. Not all persons who have sexually offended against children are attracted to minors and not all those who are attracted to minors offend. Nonetheless, all persons who have sexually offended (as well as those who are attracted to minors and have not offended) are perceived as dangerous predators.

The blame assigned to those who carry the label of "sex offender" often lacks the element of compassion and the desire to help individuals who have sexually offended. Two of the attributes necessary for compassion are common humanity and non-judgment (Gilbert and Choden 2015; Neff 2011). Gilbert and Choden (2015) noted that the human tendency toward tribalism is most likely one of the primary barriers to compassion. Those who do not share the attributes we assign to our group and ourselves are easily dismissed. They are "not like us." As a result, it is difficult to acknowledge their humanity, and it is easy to judge their actions from a non-compassionate perspective.

We do not consider those who are not like us to be our "fellow human beings, who want to be free of suffering, to be happy, to love their children and see them grow and flourish; we block from trying to see the world through their eyes" (Gilbert and Choden 2015, p. 117). Perceiving those who have committed sexual offenses as monsters who are not like us has contributed to reactive legislation (Letourneau et al. 2014) that is ineffective in increasing public safety and that promotes stigmatization, isolation, and a host of negative psychological symptoms (Lasher and McGrath 2012; Tewksbury 2012). Understanding the complexities of what contributes to desistance from sexual offending provides a framework for moving toward strategies that are both more effective in preventing sexual abuse and promote both compassion toward individuals who have sexually offended and self-compassion among those who have offended.

## 1.1. Desistance from Criminal Behavior

"A leper came to him [and kneeling down] begged him and said, 'If you wish, you can make me clean.' Moved with pity, he stretched out his hand, touched him, and said to him, 'I do will it. Be made clean.'" (Mark 1:40–45, New American Bible)

There is no consensus on the definition of desistance from criminal behavior. Definitions of desistance range from an event or a point in time when individuals report they ceased to engage in a particular behavior or set of behaviors, to an ongoing dynamic process that includes both having stopped and staying stopped (Willis et al. 2010). Two of the potential pathways that contribute to desistance are social controls and identity change. The first is informal social controls (controls not associated with formal law enforcement mechanisms such as parole and probation) resulting from external circumstances such as being employed or having an intimate partner (Sampson and Laub 1993). There is extensive research demonstrating that those who have sexually offended have difficulty attaining the external circumstances that promote informal social control (Harris et al. 2017; Jeglic et al. 2011; Levenson et al. 2007; Mercado et al. 2008). It is common for this population to be subjected to restrictions to where they can live, work, play, and even worship. Initial research examining informal social controls for this population suggests that informal social controls such as employment and being married may not impact desistance for this population to the extent they do for desistance from generalized crime (Harris 2014; Lussier and McCuish 2016). This may be related to the additional barriers to employment and relationships faced by this population.

A second contributor to desistance involves an identity change that is characterized as perceiving one's self as a changed person (Giordano et al. 2002; Maruna and Farrall 2004). Cognitive transformations related to having a changed self-perception, such as adopting a new identity as non-offending and being committed to helping others avoid offending, occur along a continuum (Harris 2014). For some of those convicted of sexual offenses, their new identity included a sense that redemption was either possible or had been achieved (Harris 2014). The concept of redemption, the feeling of being saved from wrongdoing or sin, positions religion and spirituality as a possible force in desistance for persons who have sexually offended.

## 1.2. Religiousness and Spirituality among Persons Who Have Sexually Offended

The role of religion and spirituality (r/s) in desistance from sexual offending is increasingly becoming an area of interest. Although in its infancy, this line of academic inquiry is supported by theoretical constructs (Kewley 2019), preliminary research findings supporting the potential contribution of r/s to desistance from sexual offending (Harris et al. 2017; Kewley et al. 2015; Kewley et al. 2017), and self-reports from those who have sexually offended that religion and spirituality are important in their recovery (Geary et al. 2004; Geary et al. 2006; Robbers 2009; Tewksbury and Mustaine 2009; Tewksbury and Zgoba 2010). Conversely, studies have argued that affiliation with religion can serve to promote criminal behavior when religious doctrine is used to support cognitive distortions that rationalize offending behaviors (Knabb et al. 2012; Topalli et al. 2012; Winder et al. 2018). Despite recent scholarly interest in how r/s might positively or negatively influence desistance, very little is

known about how r/s constructs present among those who have sexually offended, or how religious and spiritual struggles might contribute to, or deter from desistance.

In addition to preliminary evidence that, for some, r/s contribute to desistance from sexual offending, many of those who have sexually offended report that religion and spirituality are important to them. After controlling for personality, a survey of 195 men who had sexually offended found that both religion and spirituality predicted satisfaction with life and positive affect (Geary et al. 2004). Nearly three-fourths (74.6%) of the men in this same sample indicated that religion and spirituality were positive forces in their recovery process (Geary et al. 2006).

Robbers (2009) surveyed 153 individuals on the sex offense registry in Virginia. Most (63%) of the respondents indicated that religion was "very important." Approximately one-third of these same study participants reported they had some level of involvement in their communities with the most frequent type of community involvement being church-related. Other studies examining persons on sex offense registries have also noted the self-reported importance of religion. In two separate studies of the psychological consequences of public sex offense registries, religion (Tewksbury and Zgoba 2010) and prayer, and/or meditation (Tewksbury and Mustaine 2009) were identified by respondents to be important coping strategies for managing registry-related stressors.

Religion and spirituality have been associated with decreased criminal behavior (Baier and Wright 2001) and desistance from criminal behavior (Giordano et al. 2008; Kewley et al. 2015). Criminal justice desistance theory has also identified r/s as instrumental in the process of desistance. Desistance theory proposes that those who desist from reengaging in past criminal behaviors adopt a reconstructed self-identity that often includes a sense of empowerment originating from an external source such as another person, a divine being, organization, philosophy, or religion (Maruna 2001). This process can involve what Maruna (2001) refers to as "redemption scripts." Redemption scripts are created by desisters and are characterized by a shift in their personal narratives from one of failure and shame to one of productivity, worthiness, and meaning and purpose. This reconstructed self-identity is prevalent in desistance from criminal activity (Laws and Ward 2011). The reconstructed self-identity might be particularly challenging for persons who have sexually offended because of the additional challenges imposed by social stigma, formal and informal restrictions on their ability to access supports, and the moral implications of their behavior for themselves and others.

### 1.3. Religious and Spiritual Struggles

Although r/s have been noted to be important to those who have sexually offended, are associated with psychological benefit, and contribute to desistance from offending behavior, there has been no discussion in the literature about how religious and spiritual (r/s) struggles might present within this population or how r/s struggle might be addressed to promote both individual well-being and desistance. There are different types of religious and spiritual struggles (Exline et al. 2014; Pargament et al. 2005), and we suggest that several may be of particular relevance to persons who have sexually offended. Specifically, moral struggle, struggles of ultimate meaning, interpersonal struggle, and divine struggle may be pertinent to individuals who have sexually offended.

#### 1.3.1. Moral Struggles

Moral struggle, an intrapersonal r/s struggle, involves internal thoughts or actions related to real or perceived transgressions from moral principles and can involve intense feelings of guilt (Exline et al. 2014). Individuals who have sexually offended can have difficulty reconciling their offense with their moral principles. For example, 71 men who had sexually offended frequently reported during interviews that they believed their offense was unforgivable, even irredeemable (Harris et al. 2017). Recent qualitative desistance research has identified "neutralization" of offending behavior among those who have sexually offended as part of the process of cognitive transformation that supports a positive self-identity when one's actions and beliefs have been incongruent with that identity (Hulley 2016). Sykes and Matza (1957) identified 5 neutralization techniques: denial of

responsibility, denial of injury, denial of the victim, condemnation of condemners, and the appeal to higher loyalties. Subsequent theoretical development suggests that these neutralization techniques are universally applied cognitive responses which serve to resolve inconsistencies between one's behavior and one's beliefs (Hazani 1991; Maruna and Copes 2005). Further, the results of Hulley (2016) research suggests that neutralization contributes to the development of a pro-social identity and self-acceptance in the early phases of desistance from sexual offending by allowing the individual to negotiate stigma that would otherwise be an obstacle to reconstruction of a pro-social identity. Thus, neutralization may aid in the resolution of moral struggle experienced by those who have sexually offended.

Viewed within the context of compassion, neutralization provides a vehicle to address the suffering that accompanies widely held cultural misconceptions that those who carry the label of "sex offender" are a homogeneous group who are at high risk of reoffending, and are incapable of change (Levenson et al. 2007; Maguire and Singer 2011; Mancini and Pickett 2016; Payne et al. 2010; Sample and Bray 2006). Gilbert (2017) defines compassion as "a sensitivity to suffering in self and others with a commitment to try to alleviate and prevent it" (p. 11). This definition expands compassion to include self-compassion as well as compassion for others and is positioned as a motivation to relieve suffering, rather than an emotion arising from exposure to suffering. Already a facilitator of prosocial behavior, compassion has the potential to further enhance the function of neutralization in the negotiation of stigma and resolution of moral struggle.

### 1.3.2. Struggles of Ultimate Meaning

Another intrapersonal struggle that has implications for those who have sexually offended involves struggles with ultimate meaning, or the absence of a profoundly meaningful life. Piedmont (2010) has defined spirituality as an intrinsic, trait-based motivational force which influences human behavior aimed at creating ultimate personal meaning and purpose. Finding meaning has been noted to be a central aspect of being human (Frankl 1988). Creating meaning is an individual pursuit that seeks to frame one's life experiences within a transcendent context, as part of something bigger than oneself (Park 2013; Piedmont 2010). Making meaning often occurs within the circumstances of daily life. Frankl (1988) suggests that meaning can be found in our work, deeds, aesthetic experiences in nature and within culture, and love. Generally, persons who have been convicted of a sexual offense serve time in prison and upon release, their name, address, conviction, and other personal information are made available to the public on a searchable website. Thus, through both physical inaccessibility and social stigmatization, many who have sexually abused others have limited access to the resources and situations most often used for making meaning, including employment, housing, parks, and relationships (Burchfield and Mingus 2008; Mercado et al. 2008; Tewksbury 2012). Frankl (1988) indicates that perhaps the most important acquisition of meaning occurs for people when they are denied access to opportunities to find meaning and are able to "rise above it and grow beyond themselves" (p. 70).

Finding meaning and purpose is a prevalent theme in desistance scripts of those who desist from criminal behavior (Maruna 2001). Consistent with desistance from non-sexual crimes, finding meaning in work and relationships, and the vision of a new self, appear to be important among those who have desisted from sexual offending (McAlinden et al. 2017). A notable difference between the narratives described in general desistance literature and sexual-offense desistance narratives is perceived barriers to the actualizing their employment and relationships goals (McAlinden et al. 2017). The sexual offense desistance narratives examined by McAlinden et al. (2017) led the authors to conclude that "the agentic willingness to change on the part of individuals like those in this sample needs to be accompanied by credible social opportunities for change and a range of external situational supports to help sex offenders achieve meaningful lives" (p. 278). For this population, intrapersonal religious and spiritual struggles with meaning and purpose may be linked to the ability to have access to, and participate in, society and to be treated with basic human dignity.

### 1.3.3. Interpersonal Struggles

Interpersonal r/s struggles can arise when negative encounters with religious people or religious institutions are experienced. Both formal and informal barriers to involvement in faith communities are encountered by persons who have sexually offended. Formal barriers include supervision restrictions imposed by parole and probation or by faith community policies. These restrictions can include requiring the person to have a chaperone, sit in an assigned seat, be excluded from church events other than services, and include being banned from attending services. Faith community policies that presented barriers to attending services was a prevalent emergent theme in interviews with 71 men who had committed sex offenses (Harris et al. 2017). Harris et al. (2017) concluded that participation in a faith community can positively impact desistance and recommended that policy and regulations restricting persons who had sexually offended from attending services be reversed.

Informal barriers include ostracism from members of faith communities. One man reported, "My previous integral role within synagogue community life was entirely blotted out by my offense . . . With a couple of exceptions, no one wished to stay in touch, still less propose reintegration." (Service User 2019, p. 145). The first study to examine attitudes of community members regarding the extent to which persons who have been convicted of sexually offending should be restricted within their faith communities found that acceptance is influenced by different demographic and religious factors (Dum et al. 2020). For instance, Dum et al. (2020) found that restrictive policies toward inclusion of persons who had sexually offended were endorsed to a greater extent by those with a stronger faith, and by those who held the belief that morality is individualistic rather than originating from God. Thus, both formal and informal barriers can leave individuals feeling further isolated and rejected instead of experiencing the benefits others in faith communities often receive, such as kindness, compassion, forgiveness, acceptance, and a sense of belonging.

### 1.3.4. Divine Struggles

Struggles with the divine typically involve experiencing negative thoughts and emotions centered on beliefs about the God of one's understanding or an individual's perceived relationship with their God. For example, persons who have sexually offended may feel punished or abandoned by God, find it difficult to ask God for help given their sexual and moral transgressions, or feel unworthy of God's love. Pallas (2014) found that among 30 men residing in the community who had been convicted of sexually offending, negative religious coping, such as feeling abandoned by God, feeling punished by God, and questioning God's love for them were significantly linked to both depression and anxiety. Additionally, in a study of individuals who were incarcerated, largely for sexual crimes (31.5%) or murder (38.4%), Allen, Phillips, Roff, Cavanaugh, and Day (Allen et al. 2008) found that when inmates reported lower levels of feeling abandoned by God, they were more likely to have a greater number of daily spiritual experiences, fewer symptoms of depression, and less desire for hastened death.

Recent research has shown that a specific form of divine struggle, Religious Crisis (RC), has a unique, significant impact on psychosocial functioning. Fox and Piedmont (2020) found that a measure of RC was significantly related to feelings of depression, anxiety, and stress, independent of the personality dimensions of the Five-Factor Model (FFM). Further, structural equation modeling supported the perspective that RC was a causal agent in creating these feelings. Piedmont, Hassinger, Sherman, Sherman, and Williams (Piedmont et al. 2007) found similar results when examining RC's predictive role in the experience of characterological impairment. The feelings of unworthiness, inadequacy, and existential rejection that are characteristic of RC represent a source of psychological distress independent of mere affective dysphoria. While the personality dimension of Neuroticism is concerned with affective dysregulation, RC concerns difficulties in the overall sense of meaning and personhood that people have in relation to their entire engagement with their reality. Disruptions in their sense of worthiness indicate problems with their core sense of self and lead to feelings of fatalism and ultimate rejection (Piedmont and Wilkins 2020). The result can be feelings of profound personal emptiness, dread, and despair. In sum, previous research suggests that divine struggles, including RC,

may have adverse associations with mental health symptoms for many individuals, including those who have sexually offended.

*1.4. Present Study*

To our knowledge, the present study represents the first to examine the religious struggles of persons convicted of sexually offending. The aim of this research was to begin to describe the nature and frequency of religious struggles among persons convicted of sexually offending and describe how these struggles are connected to personality and mental health for this population. Additionally, based on the findings of previous compassion-focused research (Jazaieri et al. 2013; Neff 2011; Matos et al. 2018), we explore associations between self-compassion, compassion for others, personality, and mental health in this sample of males convicted of sexually offending.

**2. Method**

*2.1. Participants*

Table 1 shows the demographics of the participants. As Table 1 indicates, White and Black participants were nearly evenly represented in the study. Participants' ages ranged from 21 to 72 years, with an average age of 51.8 years. Despite nearly all participants (92.8%) having acquired either some college or a degree following high school, nearly 50% were either unemployed or employed part-time, and 70.0% earned $20,000 per year or less. The majority of the participants (56.7%) lived with either family or friends, reported they were single and not dating (56.7%), and were affiliated with Christian faith traditions (75.9%). Participant registration, offense, and victim information are shown in Table 2.

**Table 1.** Participant demographics.

| Demographic | *N* | % | Demographic | *N* | % |
|---|---|---|---|---|---|
| Race | | | Age of Participant [†] | | |
| Caucasian | 13 | 43.3 | 18–21 | 1 | 3.3 |
| Black | 12 | 40.0 | 29–34 | 3 | 10.0 |
| Other | 4 | 13/3 | 35–49 | 6 | 20.0 |
| Asian | 1 | 3.3 | 50–59 | 10 | 33.3 |
| Total | 30 | 100 | 60–69 | 8 | 26.7 |
| Level of Education | | | Residence | | |
| Some college | 12 | 42.9 | Lives with family | 12 | 40.0 |
| Bachelor's degree | 10 | 35.7 | Lives alone | 11 | 36.7 |
| High School/GED | 2 | 7.1 | Lives with friends | 5 | 16.7 |
| A.A. degree | 2 | 7.1 | Residential program | 1 | 3.3 |
| Graduate degree | 2 | 7.1 | Other | 1 | 3.3 |
| Total [†] | 28 | 100 | Total | 30 | 100 |
| Relationship Status | | | Religious Affiliation | | |
| Single not dating | 17 | 56.7 | Protestant | 17 | 58.6 |
| Married or partnered | 9 | 30.0 | Catholic | 5 | 17.2 |
| Dating one person | 4 | 13.3 | Atheist/Agnostic | 4 | 13.8 |
| Other | 0 | 0 | Other faith | 3 | 10.3 |
| Total | 30 | 100 | Total [†] | 29 | 100 |
| Employment Status | | | Annual Income | | |
| Unemployed | 12 | 41.4 | <$10,000 | 14 | 46.7 |
| Retired | 10 | 34.5 | $10,000–$20,000 | 7 | 23.3 |
| Full time | 5 | 17.2 | $20,000–$30,000 | 4 | 13.3 |
| Part time | 2 | 6.9 | $30,000–$40,000 | 3 | 10.0 |
| Total [†] | 29 | 100 | $50,000–$60,000 | 1 | 3.3 |
| | | | $60,000–$70,000 | 1 | 3.3 |
| | | | Total | 30 | 100 |

Note: GED = General Education Diploma. [†] $N \neq 30$ due to missing data.

**Table 2.** Participant registration, sexual offense, and victim information.

| Category | N | Valid % | Category | N | Valid % |
|---|---|---|---|---|---|
| Registration Period | | | Victim Involved in Offense | | |
|    10 years or less | 3 | 10.3 |    Contact victim | 23 | 76.7 |
|    15 years | 3 | 10.3 |    Non-contact victim(s) | 7 | 23.3 |
|    25 years | 3 | 10.3 |    Total | 30 | 100.0 |
|    Lifetime | 20 | 69.0 | | | |
|    Total [†] | 29 | 100.0 | | | |
| Years on the Registry | | | Victim Years of Age | | |
|    1 to 2 years | 5 | 16.7 |    5 to 10 | 8 | 27.6 |
|    2 to 5 years | 5 | 16.7 |    13 to 17 | 13 | 44.8 |
|    5 to 10 years | 10 | 33.3 |    $\geq 18$ | 1 | 3.5 |
|    10 to 15 years | 10 | 33.3 |    Unknown * | 7 | 24.1 |
|    Total | 30 | 100.0 |    Total [†] | 29 | 100.0 |
| Age at Time of First Offense | | | Sex of Victim—First Offense | | |
|    19 to 25 | 11 | 36.7 |    Female | 15 | 50.0 |
|    26 to 40 | 5 | 16.6 |    Male | 8 | 26.7 |
|    41 to 48 | 8 | 26.7 |    Unknown * | 7 | 23.3 |
|    54 to 69 | 6 | 20.0 |    Total | 30 | 100.0 |
|    Total | 30 | 100.0 | | | |
| Relationship to Victim | | | Multiple Offenses | | |
|    Family member | 8 | 27.6 |    Yes | 5 | 17.9 |
|    Friend of mine | 6 | 20.7 |    No | 23 | 82.1 |
|    Friend of family | 4 | 13.8 |    Total [†] | 28 | 100.0 |
|    Stranger | 4 | 13.8 | | | |
|    Online victim | 7 | 24.1 | | | |
|    Total [†] | 29 | 100.0 | | | |

Note. * Age and sex of victim(s) are unknown because these offenses involved online victims. Unknown ages represent victims under 18 years of age. [†] $N \neq 30$ due to missing data.

Most participants were required to register for life and had been registering for between 5 to 15 years. On average, participants had been registering for just under 8 years. The average age of participants at the time of their first offense was 38 years old. As is typical of sexual offending, the victims of most participants were family members, or friends of family or themselves. Nearly one-quarter of participants were convicted of viewing sexually explicit images of children. The average age of the victim of those whose offenses that did involve "hands on" victims was 13 years old, and the victims were most often female. The majority of participants reported having one offense.

## 2.2. Procedure

Recruitment of participants for this study entailed sending a letter inviting 1858 men on the sex offense registry in a major city and the surrounding county of a Mid-Atlantic state to participate in the study. A total of 73 recruitment letters were returned as undeliverable. The study sought 50 participants for a compassion-focused treatment intervention study; however, only 30 (1.7% of the 1858) participants consented and completed questionnaires. The intent of the present paper is to describe participants' religiousness, mental health, and level of self and other compassion prior to receiving intervention. A separate manuscript reporting the procedures, protocol, and results of the compassion-focused intervention is in preparation for publication.

All participants were males over the age of 18, who had been required to register on the state's sex offense registry for a minimum of 12 months. Participants were required to have a high school diploma or a general education diploma. All study procedures were approved by the Loyola University Maryland Institutional Review Board. Examination of frequency distributions for each of the scales revealed that several of the survey instruments had missing responses. Missing values were replaced

using an SPSS syntax program designed to replace items using the mean of at least 80% of valid scale responses.

*2.3. Measures*

### 2.3.1. Assessment of Spirituality and Religious Sentiments (ASPIRES)

Developed by Piedmont (2010), this 35-item scale captures spirituality and religiousness from a motivational, non-denominational perspective. The instrument is comprised of two domains: Religious Sentiments (RS) and Spiritual Transcendence (ST). The author defined RS as attitudes about religion that develop within social, cultural, and educational contexts. The measure of RS is comprised of 12 Likert-type items with responses that range from 1 (Strongly Disagree) to 5 (Strongly Agree) and includes two subscales, Religious Crisis (RC) and Religious Involvement (RI). The four-item RC subscale, which identifies difficulties individuals may be experiencing relative to the God of their understanding and to their faith community was used in this study as an indicator of spiritual struggle. The eight item RI scale examines the extent to which individuals are actively engaged in religious rituals and activities. Both RC and RI capture the value individuals place on their involvement in religious activities. Piedmont (2010) reported good self and observer alpha reliabilities for scores on RC as 0.78 (self) and 0.81 (observer), with a cross-observer convergence of $r = 0.34$ (2998, 981), $p < 0.001$.

The ST scale contains 23 items and are captured on a 5-point Likert-type scale with responses that range from 1 (Strongly Disagree) to 5 (Strongly Agree). ST measures an intrinsic, trait-based motivational force which influences human behavior aimed at creating ultimate personal meaning and purpose (Piedmont 2010). The three correlated facets that comprise ST are Prayer Fulfillment (PF), Universality (UN), and Connectedness (CN). Piedmont (2010) reported acceptable alpha reliabilities for total ST, and for each of the facets. Normative alpha reliabilities for ST scores are 0.93 (self) and 0.90 (observer), with cross-observer convergence between the two alphas of 0.57.

### 2.3.2. Self-Compassion Scale (SCS)

Developed by Neff (2003), the SCS is a 26-item Likert-type scale with ratings from 1 (Almost Never) to 5 (Almost Always). Neff (2003) defined self-compassion as being open to one's own suffering while extending kindness to self, along with the desire to alleviate one's own suffering. The SCS has six subscales, (Self-Kindness, Self-Judgment, Common Humanity, Isolation, Mindfulness, and Over-Identification). Neff (2011) research showed that sub-scale scores can be combined to create a total self-compassion score, with an alpha reliability for the total score of 0.92. The present study administered the 12-item SCS short form and therefore utilized the total self-compassion score. Raes, Pommier, Neff, and Van Gucht (Raes et al. 2011) found nearly identical correlations between the short-form SCS and the long-form SCS. Short-form example items include, "I try to see my failings as part of the human condition," and "I'm disapproving and judgmental about my own flaws and inadequacies" (reversed scored).

### 2.3.3. Compassion for Others Scale (CS)

Developed by Pommier (2011), the CS is a 24-item unidimensional Likert-type scale measuring the frequency of compassion for others. Building upon Neff (2003) model of self-compassion, the CS conceptualizes compassion from a Buddhist perspective, viewing compassion as a source of motivation to alleviate the suffering of others. Responses to the CS are rated from 1 (Almost Never) to 5 (Almost Always). Pommier (2011) found that CS scores demonstrated good reliability with both Cronbach's alpha and split-half coefficients of 0.90 (Pommier 2011). Example items include, "If I see someone going through a difficult time, I try to be caring toward that person," and "sometimes when people talk about their problems, I feel like I don't care" (reversed scored).

### 2.3.4. Other as Shamer Scale (OAS)

Developed by Gilbert et al. (as cited in Allan et al. 1994; Goss et al. 1994), the OAS is a measure of external shame, the experience of shame resulting from individual beliefs about how others perceive them. The OAS is an 18-item Likert-type self-report scale with responses ranging from 0 (Never) to 4 (Almost Always). The OAS includes three subscales (and one item that is only included when adding up the total OAS score for all 18 items): Inferiority (seven items, i.e., "I feel insecure about others' opinion of me"), Emptiness (four items, i.e., "others see me as empty and unfulfilled"), and Mistake (six items, i.e., "others are critical and punishing when I make a mistake"). Each of the subscales have demonstrated good alpha reliability, ranging from 0.69 for emptiness to 0.80 for inferiority (Balsamo et al. 2015). The OAS scores have been shown to have good internal consistency with an alpha of 0.92. Test-retest reliability following a 10-day interval was found to be excellent ($r = 0.82$, $p < 0.01$) according to Balsamo et al. (2015).

### 2.3.5. Perceived Stress Scale-4 (PSS-4)

Developed by Cohen, Kamarck, and Mermelstein (Cohen et al. 1983), participants responded to four questions about their feelings and thoughts during the last month with response choices ranging from 0 (Never) to 4 (Very Often). Items on the PSS-4 are "In the last month, how often have you felt that you were unable to control the important things in your life," "In the last month, how often have you felt confident about your ability to handle your personal problems?" (reverse scored), "In the last month, how often have you felt that things were going your way?" (reverse scored), and "In the last month, how often have you felt difficulties were piling up so high that you could not overcome them?" Higher scores indicate greater perceived stress. The PSS-4 has been used widely and has adequate alpha reliability as demonstrated in a U.S. sample of 2136 adults at initial assessment ($\alpha = 0.72$) and three years later ($\alpha = 0.77$).

### 2.3.6. The Hospital Anxiety and Depression Scale (HADS)

The HADS (Zigmond and Snaith 1983) is a 14-item scale designed to measure symptoms of anxiety and depression. Seven of the HADS items are focused on symptoms of anxiety (HADS-A, i.e., "worrying thoughts go through my mind") with the remaining seven items focused on symptoms of depression (HADS-D, i.e., "I feel as if I'm slowed down"). All items are rated on a 4-item scale ranging from 0 (Least Amount of Anxiety or Depression) to 3 (Most Anxiety or Depression). The HADS has been widely used with both hospital patients and with nonclinical populations. In a review of 747 studies that used the HADS, 15 studies reported alpha reliabilities for scores ranging between 0.67 to 0.90 (mean = 0.82) for the HADS-D and between 0.68 and 0.93 (mean = 0.83) for the HADS-A (Bjelland et al. 2002).

### 2.3.7. Beck Hopelessness Scale (BHS)

Developed by Beck and his colleagues, the BHS (Beck et al. 1974) is a 20-item true or false instrument. Each item is scored either 0 for False, or 1 for True (i.e., "the future seems vague and uncertain to me"). Item scores are summed for a total hopelessness score. Scale item inter-correlations were significant and ranged from 0.39 to 0.76. Alpha reliability of the BHS was found to be 0.93. A similar result was found in a recent sample of persons convicted of sexual offenses, where the Cronbach's alpha was 0.95 (Jeglic et al. 2011). Factor analysis identified a three-factor solution for the BHS, Feelings about the Future, Loss of Motivation, and Future Expectations.

### 2.3.8. International Personality Item Pool–50 (IPIP-50)

The IPIP-50 is a 50-item measure of the Five Factor Model (FFM) personality dimensions of Emotional Stability (Neuroticism), Extraversion, Intellect/Imagination (Openness), Agreeableness, and Conscientiousness. The IPIP-50 is comprised of 10 items for each of the personality dimension

subscales. The items are rated on a 5-point scale ranging from 1 (Very Inaccurate) to 5 (Very Accurate). The IPIP-50 has been used frequently, and alpha reliabilities have been consistent across study populations and in a wide range of cultural environments (Gow et al. 2005; Lim and Ployhart 2006; Zheng et al. 2008; Thalmayer et al. 2001). Scores on the IPIP-50 have compared well with other measures of the FFM (Gow et al. 2005; Lim and Ployhart 2006), resulting in comparable alpha reliabilities (Emotional Stability = 0.87, Extraversion = 0.80, Intellect/Imagination = 0.81, Agreeableness = 0.80, and Conscientiousness = 0.79; Mlacic and Goldberg 2007). The FFM scale labels are used in the remainder of this paper for ease of understanding. Emotional Stability is referred to as Neuroticism and Intellect/Imagination is referred to as Openness. Emotional Stability items were reverse coded to coincide with the label of Neuroticism.

### 2.3.9. Demographic Questionnaire

Demographic information requested included age, race/ethnicity, religious affiliation, residential status, relationship status, employment status, educational level, income, length of time on registry, length of time required to register, sex and age of first victim, and relationship to victim.

## 3. Results

### 3.1. Descriptive Statistics

Means, standard deviations (SD), ranges, and alpha reliabilities for scales are displayed in Table 3. ASPIRES subscale scores are presented as T-scores with a mean = 50, and SD = 10, based on normative data (Piedmont 2010). T-scores between 45 and 55 are considered normative. As can be seen, participants scored below average on all the ST facet scales and total score while also scoring above average on Religious Crisis. One sample *t*-tests were used to compare means on the ASPIRES in this sample with the normative group. Results demonstrate that on constructive attitudes of spirituality that develop within social, cultural, and educational contexts (prayer fulfillment ($t = -9.32$, $p < 0.001$), universality ($t = -7.75$, $p < 0.001$), connectedness ($t = -4.20$, $p < 0.001$), and total spiritual transcendence ($t = -8.65$, $p < 0.001$)), mean scores were significantly lower than the normative sample. Conversely, religious crisis mean scores ($t = 4.91$, $p < 0.001$) fell significantly above the mean of the normative group, indicating that the men in this sample experienced greater difficulties relative to the God of their understanding and their faith community. Level of distress for anxiety, depression, and hopelessness are determined by comparison of scale scores to established cut-off scores. The mean HADS-A scale score was 8.73 (SD = 4.53), which places this sample in the "mild" category for anxiety (scores ranging 8–10), while the mean HADS-D scale scores was 5.57 (SD = 3.47), indicating "normal" levels of low mood (scores ranging 0–7; Snaith and Zigmond 1994). The mean BHS scale score was 4.47 (SD 5.11), which falls into the "mild hopelessness" category (Beck and Steer 1988). Mean levels of perceived stress were 7.17 (SD = 3.9) in this sample, indicating higher levels of perceived stress than normative scores on the PSS-4 (M = 6.11) found by Wartting et al. (2013).

Mean self-compassion scale scores for this sample were higher, 3.18 (SD = 0.83), than mean scores for a clinical sample of college males, 2.88 (SD = 0.71; Lockard et al. 2014). Similarly, compassion for other mean levels of 4.00 (SD = 0.55) were higher for the current sample than the mean of 3.73 (SD = 0.71) for an adult community sample of males (Pommier et al. 2020). Finally, mean scale scores for perceiving others as shaming, M = 32.70 (SD = 19.09) were high relative to the mean of 20.0 (SD = 10.1) reported by Goss et al. (1994), indicating that experiences of shame resulting from beliefs about how they believed others perceive them were comparatively high. Concerning scores on the FFM personality domains measured by the IPIP-50, these values were compared with those presented by Robertson, Janga, Piedmont, Sherman, and Williams (Robertson et al. 2017). The current sample scored lower on Neuroticism and Extraversion, with the mean score for Neuroticism of 29.01 (SD = 9.52) for the current sample compared to 32.25 (SD = 6.89), and the mean score for Extraversion of 28.67 (SD = 8.94) for the current sample, compared to 31.74 (SD = 6.93) for the comparison sample.

This sample scored higher on Conscientiousness with a mean of 37.36 (SD = 7.59) compared to 35.63 (SD = 5.09) and Openness with a mean of 38.00 (SD = 6.58), compared to 36.33 (SD = 5.00). Levels of Agreeableness were nearly identical with the current sample mean of 38.36 (SD = 7.60), compared with a mean of 38.35 (SD = 5.31).

**Table 3.** Means (M), standard deviations (SD), ranges and alpha reliabilities (α) for study variables.

| Variable | M | SD | Possible Range | α |
|---|---|---|---|---|
| ASPIRES * | | | | |
| Prayer Fulfillment | 42.70 [†] | 4.29 | N/A | 0.85 |
| Universality | 41.00 [†] | 6.36 | N/A | 0.65 |
| Connectedness | 42.89 [†] | 9.28 | N/A | 0.54 |
| Spiritual Transcendence | 41.01 [†] | 5.69 | N/A | 0.85 |
| Religious Struggle | 57.95 [†] | 8.87 | N/A | 0.27 |
| Religious Involvement | 47.60 [†] | 9.58 | N/A | 0.86 |
| Personality | | | | |
| Neuroticism | 29.01 | 9.52 | 1–50 | 0.87 |
| Extroversion | 28.67 | 8.94 | 1–50 | 0.86 |
| Openness | 38.00 | 6.58 | 1–50 | 0.81 |
| Agreeableness | 38.36 | 7.60 | 1–50 | 0.80 |
| Conscientiousness | 37.36 | 7.59 | 1–50 | 0.79 |
| Perceived Stress | 7.17 | 3.19 | 0–16 | 0.75 |
| Anxiety | 8.73 | 4.53 | 0–21 | 0.85 |
| Depression | 5.57 | 3.47 | 0–21 | 0.70 |
| Hopelessness | 4.47 | 5.11 | 0–20 | 0.91 |
| Self-Compassion | 3.18 | 0.83 | 1–130 | 0.85 |
| Compassion for Others | 4.00 | 0.55 | 1–120 | 0.86 |
| Other as Shamer | 32.70 | 19.09 | 0–72 | 0.97 |

Note. $N$ = 30. ASPIRES = Assessment of Spirituality and Religious Sentiments * = ASPIRES subscale scores are presented as T-scores with a mean = 50, and SD = 10, based on normative data (Piedmont 2010). [†] = sample mean is significantly different from the normative group mean at the $p < 0.001$ level.

### 3.2. Associations between Religion, Compassion, Mental Health, and Personality

Table 4 displays the bivariate correlations between study variables. Greater levels of self-reported spiritual transcendence were significantly related to lower ratings of perceived stress ($r = -0.39$) and lower levels of anxiety ($r = -0.40$). Religious involvement was significantly inversely associated with both hopelessness ($r = -0.36$) and religious crisis ($r = -0.54$). Higher levels of self-reported religious struggle (as measured by the religious crisis subscale of the ASPIRES) were significantly associated with greater levels of perceived stress ($r = 0.52$), depression ($r = 0.53$), anxiety ($r = 0.66$), hopelessness ($r = 0.56$), and shame ($r = 0.66$). Additionally, greater religious struggle was significantly associated with less self-compassion ($r = -0.54$). Finally, greater self-compassion was significantly related to lower levels of perceived stress ($r = -0.47$), shame ($r = -0.43$), depression ($r = -62$), anxiety ($r = -0.57$), hopelessness ($r = -0.58$), and religious struggle ($r = -0.54$).

Points of interest pertaining to the FFM personality dimensions can also be seen in Table 4. Neuroticism was significantly correlated with all mental health measures: shame ($r = 0.75$), stress ($r = 0.59$), depression ($r = 0.58$), anxiety ($0.75$), and hopelessness ($r = 0.59$). Neuroticism was also significantly positively correlated with religious crisis ($r = 0.76$) and negatively correlated with self-compassion ($r = -0.50$). However, Neuroticism was not significantly linked to religious involvement, spiritual transcendence, or compassion for others scores. Openness was significantly positively linked to perceived stress ($r = 0.39$) and negatively associated with self-compassion ($r = -0.37$). Agreeableness was significantly associated with compassion for others ($r = 0.52$) and Conscientiousness was inversely correlated with experiencing others as shaming ($r = -0.38$).

**Table 4.** Correlations between religious struggle, self and other compassion, personality, and mental health variables.

|     | Sh | St | D | An | H | N | E | O | A | C | RI | RC | ST | SC |
|-----|-----|-----|-----|-----|-----|-----|-----|-----|-----|-----|-----|-----|-----|-----|
| N | 0.75 *** | 0.59 ** | 0.58 ** | 0.75 *** | 0.59 ** | — | | | | | | | | |
| E | 0.36 * | 0.23 | 0.32 | 0.07 | 0.51 ** | 0.47 ** | — | | | | | | | |
| O | 0.15 | 0.39 * | 0.20 | 0.34 | 0.05 | 0.22 | −0.13 | — | | | | | | |
| A | 0.05 | 25 | 0.25 | 0.23 | −0.03 | 0.20 | −0.08 | 0.71 *** | — | | | | | |
| C | −0.38 * | −0.11 | −0.22 | −0.24 | −0.35 | −0.08 | −0.05 | 0.33 | 0.46 * | — | | | | |
| RI | −0.23 | −0.20 | −0.31 | −0.23 | −0.36 * | −0.16 | −0.26 | −0.01 | 0.02 | 0.09 | — | | | |
| RC | 0.66 *** | 0.52 ** | 0.53 ** | 0.66 *** | 0.56 ** | 0.76 *** | 0.32 | 0.14 | 0.01 | −0.27 | −0.54 ** | — | | |
| ST | −0.34 | −0.39 * | −0.12 | −0.40 * | −0.09 | −0.32 | −0.16 | −0.01 | −0.04 | 0.21 | 0.04 | −0.26 | — | |
| SC | −0.43 * | −0.47 * | −0.62 *** | −0.57 ** | −0.58 ** | −0.50 ** | −0.27 | −0.37 * | −0.21 | 0.26 | 0.17 | −0.54 * | 0.23 | — |
| CO | −0.04 | 0.07 | 0.01 | 0.11 | −0.09 | −0.14 | −0.38 * | 0.25 | 0.52 ** | −0.07 | 0.04 | −0.11 | 0.05 | 0.05 |

Note: * $p < 0.05$; ** $p < 0.01$; *** $p < 0.001$. $N = 30$. Sh = Shame (Other as Shamer Scale); St = Stress; D = Depression; H = Hopelessness; An = Anxiety; N = Neuroticism; E = Extraversion; O = Openness; A = Agreeableness; C = Conscientiousness; RI = Religious Involvement; RC = Religious Crisis; ST = Spiritual Transcendence; SC = Self-Compassion; CO = Compassion for others.

## 4. Discussion

The literature on desistance from crime supports the utility of involvement in faith communities as a positive force in assisting those convicted of crimes to return to their communities and remain crime free (Baier and Wright 2001; McCullough and Willoughby 2009). However, little attention has been paid to the religious struggles that individuals who have sexually offended may face in their lives on the road to recovery/desistance from offending. The findings from this study inform the ways in which religion and spiritualty, personality, and self and other compassion are linked to the mental health of persons who have sexually offended, which in turn are tied to factors in desistance from sexually offending (Morley et al. 2016). More specifically, this study was the first to examine empirically the religious and spiritual struggles of men who have sexually offended and found that RC, a combination of divine and interpersonal struggle, was experienced at significantly higher levels than in the normative group (Piedmont 2010). At the same time, the men in this study scored significantly lower than the normative group (Piedmont 2010) on a measure of Spiritual Transcendence.

Higher scores on the Religious Crisis scale indicate the experience of more divine struggle related to feeling punished or abandoned by God, as well as feeling unwilling or unable to work collaboratively with God to make life decisions. Given the significant life challenges individuals who have sexually offended face in their lives, such as struggles with taboo sexual preferences, moral decision making, judgement from others, social rejection, and isolation, it is understandable that these individuals could feel they are being reprimanded or forgotten by the God of their understanding for their life choices and circumstances. Additionally, individuals who have sexually offended may find it difficult to talk to or rely on God for guidance because they may view God as failing to help them in previous situations and/or permitting the undesirable and trying circumstances in their lives. RC also measures feelings of isolation in one's faith community, thereby tapping into interpersonal religious and spiritual struggles. Consequently, higher levels of religious struggle in this sample might also be related to social rejection and evaluation emanating from religious individuals and religious communities themselves.

Moreover, based on the ASPIRES model developed by Piedmont (Piedmont 2010; Piedmont and Wilkins 2020), these findings suggest that high scores on religious crisis indicate these participants may be experiencing a disrupted relationship to the transcendent that is creating emotional distress. As shown by previous research, such high scores may also indicate the presence of an existential crisis of worthiness (Piedmont and Wilkins 2020), including possible bitterness and dissatisfaction, as well as suspiciousness of the motivations of others (Piedmont 2010). High RC scores may also indicate higher levels of personal affective distress, a poor sense of life satisfaction, lowered self-esteem, and lowered psychological maturity (Fox and Piedmont 2020). Thus, the existing body of research on RC helps conceptualize the types of religious, spiritual, emotional, and behavioral challenges that individuals who have sexually offended routinely face and provides insight into potentially useful areas for intervention (discussed in detail later in the paper).

Regarding low scores on spirituality as measured with the STS, Piedmont (2010) argued that these individuals may place a greater focus on the tangible realities of daily living, whereby personal concerns and issues are of greater concern than consideration of spiritual realities that lie beyond this physical universe. Such scores indicate a more self-oriented focus and possible difficulty understanding failure and disappointment in life (Piedmont 2004). This profile makes sense considering that individuals who have been on the sexual offense registry are dealing with the immediate stressors of finding housing, employment, and regaining the acceptance of family, friends, and community. Thus, it is likely in their best interest to focus on practical, tangible personal and spiritual issues. Notably, religious involvement (RI) was significantly linked to lower levels of hopelessness and religious crisis. These findings are consistent with previous research suggesting that engagement in religious activities, fostering a relationship with the God of one's understanding, and participating in faith communities may be associated with benefits for individuals who have sexually offended (Geary et al. 2004, 2006).

Importantly, the influence of lower levels of spirituality and higher levels of religious crisis on psychological and spiritual factors also detract from components known to contribute to desistance. For instance, several components of Maruna (2001) redemption scripts align with the concept of spiritual transcendence. Perhaps most notably is becoming aware of one's intrinsic goodness, creating meaning and purpose from one's past experiences, and being motivated to contribute to the larger society, often with a focus on giving to future generations in the form of helping them to avoid or desist from offending. Engagement in faith communities provides persons who have sexually offended with the prospect of seeking both forgiveness (Blagden et al. 2020) and psychological comfort (Kewley et al. 2017). Their religious involvement also enables them to signal to other people, or to God, that as a result of a religious experience of redemption and forgiveness, they have changed.

### 4.1. Relationship between R/S Struggle, Mental Health, and Personality

As the first study of r/s struggles among men who have sexually offended, a primary objective of this study was to assess how religion and spirituality were related to mental health among participants. The psychological symptoms related to spirituality as assessed by the STS were lower perceived stress and anxiety. Although mean scores for this sample are significantly below the normative group, the associations with lower perceived stress and anxiety may be an indication that the men in this sample are benefiting, at least somewhat, from universal mental health benefits commonly attributed to r/s in other populations. However, possible explanations for the generally lower than average STS scores might be attributed to the limits placed on the extent of involvement permitted for these men within their faith communities. Often restricted from full participation, access to social support can be difficult as well as anxiety provoking. Social support has been identified as a mediator of the link between r/s and mental health (Holt et al. 2018). A 5-year longitudinal study examining the mediation effect of social support between religious beliefs and behaviors and a range of mental and physical health outcomes among African-American men found that negative interactions with congregational members predicted increases in depressive symptoms and decreases in emotional functioning (Holt et al. 2018). Another possible explanation for lower than average STS scores for this sample may be that these men lack a sense of meaning and purpose, which is related to mental health and well-being (Park 2013). Further support for a possible lack of meaning and purpose for the men in this sample is evidenced by the finding that participants in this study tended to score higher on neuroticism, and higher on religious crisis. This indicates an interesting profile, suggesting that the distress being experienced by these individuals may be a result of two independent psychological forces (numinous and personality-based) combining to create feelings of emotional dysphoria (Fox and Piedmont 2020).

Despite the small sample size, religious struggle was found to be high among these men and was strongly associated with lower mental health scores for stress, shame, depression, anxiety, and hopelessness. Religious struggle and the related psychological symptomology are not entirely surprising given the obstacles to community reintegration faced by this population. It is not uncommon

for this population to be isolated and rejected by landlords, employers, and family and friends, as the result of formal sanctions imposed by governmental and organizational law and policy (Levenson and Cotter 2005; Levenson et al. 2007), by informal social controls in the form of disapproval and harassment from the community, and by self-imposed isolation as a response to stigmatization (Burchfield and Mingus 2008; Mercado et al. 2008; Tewksbury 2005). Rejection by persons of faith and faith communities can also be a reality for persons who have sexually offended (Dum et al. 2020).

### 4.2. Relationship between Religious Struggle, Mental Health, and Compassion

A relatively strong inverse relationship was found between religious struggle and self-compassion, suggesting that self-compassion may serve to lessen religious struggle. Self-compassion indicates there is an awareness of one's own suffering and a motivation to relieve that suffering. The motivational force to relieve suffering may allow space for the acknowledgement that one has not been abandoned by God and that God is an accessible source for support in managing life, thus minimizing religious struggle in the face of adversity. Finding, recognizing, and accepting oneself as being good despite being flawed, weak, and broken is at the heart of self-compassion and may be a pathway to finding a personal sense of existential worthiness.

The results from this sample support previous findings that self-compassion is negatively related to shame (Woods and Proeve 2014), stress (Hall et al. 2013), depression and anxiety (Baker et al. 2019; Frostadottir and Dorjee 2019; Sommers-Spijkerman et al. 2018), and hopelessness (Chu et al. 2018). Thus, self-compassion may be an important factor in improving the mental health of this population given its strong inverse correlations with symptoms of mental distress and religious struggle. Increasing self-compassion may also have positive implications for deterring reoffending. Morley et al. (2016) found that self-compassion is associated with predictors of accepted criminogenic factors. In other words, self-compassion appeared to be linked to many of the characteristics, situations, and life challenges that have been found to impact risk of criminal behavior. Specifically, the authors examined a convenience sample of incarcerated persons and found significant positive associations between self-compassion and self-control, social connectedness, and self-esteem. Significant negative associations were found between self-compassion and impulsivity, risk seeking behavior, self-centeredness, and temper. Morley et al. (2016) concluded that self-compassion is negatively related to predictors of criminal behavior. Ultimately, improvements in self-compassion for this population show promise for increasing well-being and concurrently decreasing negative psychological symptomology. The resulting psychological trajectory may better prepare individuals who have sexually offended for more successful community integration and serve as a protective factor for reoffending.

### 4.3. Compassion in Clinical Treatment and Faith Communities

The results of this study provide preliminary support for inclusion of spiritually-informed approaches to therapy with this population. Clinical approaches that seek to promote an enriched sense of meaning and address religious struggle should be considered.

Traditionally, treatment for persons who have sexually offended have focused on minimizing factors that increase risk of reoffending, often to the exclusion of other factors important in creating a meaningful life. The Good Lives Model (GLM) is considered a strengths-based approach and focuses on eleven primary essential goods that are sought out by all humans (Yates and Ward 2008). Notably, one of the essential human goods identified in the GLM is spiritualty. Therapeutic approaches that promote the cultivation of compassion may be particularly well suited to enhance spirituality, mitigate religious struggle, and contribute to desistance. Recently, approaches to therapy such as Acceptance and Commitment Therapy (ACT) and Compassion-Focused Therapy (CFT) have been examined within the framework for the GLM and proposed as interventions that can more effectively treat persons who have sexually offended (Walton and Hocken 2019). Compassion- and acceptance-oriented approaches show promise in treatment of persons who have sexually offended in that they (a) support

functional relationships with paraphilic thoughts, (b) promote a shift from identities rife with shame and internal avoidance to identities that encourage self-acceptance, (c) build on personal strengths to acquire the essential human goods identified in the GLM in prosocial ways rather than focus solely on risk reduction, and (d) are trauma-sensitive (Walton and Hocken 2019).

Enhancing capacity for compassion among members of faith communities may also be an important consideration. Rather than focusing solely on transforming self-identities of those who have offended, it can be argued that expanding the transformative process to promote r/s for all congregants, those who have and have not offended, would ultimately contribute to religious and spiritual growth for all. Compassion is a central tenet of all major faith traditions. For instance, the desert mothers and fathers developed compassion through awareness of their own human frailties, recognizing that knowledge of their own weaknesses connected them with all of humankind (Kurtz and Ketcham 1992). Among the various faith traditions, compassion is integral to Buddhist philosophy. Current compassion-based therapies are rooted in the Buddhist approach to compassion. Understanding of compassion within compassion-based approaches to therapy stress the awareness of suffering of self and others, the desire to relieve and prevent that suffering, and the ability to accept the self and others without judgment when pain is present (Germer 2009; Gilbert 2017; Jazaieri et al. 2013; Neff 2011).

We foster compassion through intentionally increasing our understanding that factors and circumstances outside of our control influence our development, thereby improving awareness of our own weaknesses and acknowledging the innate human potential for each of us to exhibit both the best and the worst of behaviors. Our sense of connectedness with others is intensified and our judgments of others are softened. Compassion for those who have offended does not mean that we are accepting of the abusive treatment of others or that those who harm others should not be punished. Rather, the wisdom inherent in compassion seeks not to harm others from a vengeful stance but to understand the suffering of the offender, sincerely wish for that person's suffering to be alleviated, and still take actions to curtail the abusive behaviors (Gilbert and Choden 2015). From a position of compassionate wisdom, the actions taken to address the harmful behaviors of offenders would seek appropriate punishment for the crime and appropriate supports to remedy the causes of suffering that lead to harmful behaviors.

*4.4. Limitations*

The small sample size for this study was less than ideal, causing intrinsically underpowered statistical results; thus, the results must be interpreted with caution. The need for replication is essential here. Second, participants self-selected to participate in this study and are likely not representative of the larger population of persons convicted of sexually offending. Those who participated may represent individuals who are more highly motivated and perhaps less intimidated than those who chose not to participate. For instance, discussion among participants (during the treatment phase of the study) included comments regarding hopefulness that their participation would help them address emotional and interpersonal challenges resulting from public registration. In addition, some participants mentioned an initial reticence to participate due to fear and mistrust of interactions related to their status as "sex offenders."

An additional limitation of the present research was the low alpha reliability found for Religious Crisis. Reasons for the low reliability may be related to a lack of a cohesive understanding of the meaning of the RC facet scale items, thus accounting for the absence of internal consistency for RC scores. Additional considerations for the poor alpha reliability for RC might reflect an internal conflict about their relationship with God. Such a conflict might be indicative of the expressed importance of r/s among many who have been convicted of sexually offending (Geary et al. 2004, 2006; Robbers 2009; Tewksbury and Mustaine 2009; Tewksbury and Zgoba 2010) and formal and informal restrictions they face to participating at all or fully within their faith traditions (Dum et al. 2020; Harris et al. 2017; Service User 2019). Another potential reason for low internal consistency reliability for RC in this sample could be due to the different types of r/s struggle measured by scale items (i.e., divine struggle, difficulty communicating with God, and interpersonal struggle). In support of this possibility, men

in this sample reported much higher mean scores on the single item measuring feelings of isolation from others in their faith community. Reliability analyses indicated that the alpha would increase substantially if this item was deleted from the scale ($\alpha = 0.40$), suggesting the scale may be tapping into different types of r/s struggles that individuals in this sample may not be experiencing similarly. Finally, the analyses were limited to descriptive and correlational examinations of the data, providing only a first look at possible characteristics of those who have offended and potential associations between r/s, mental health, compassion, and personality.

### 4.5. Future Research

This study represents the first known look at religious struggles among a sample of men convicted of sexually offending. Given the evidence pointing to the importance of r/s for some individuals who have sexually offended, the psychological and social support benefits of r/s communities, that spirituality is a component of the GLM, and the interest in the application of spiritually-informed therapeutic approaches for this population, additional research examining the extent and types of religious struggles for this population should be pursued. Scales to consider for use in future research with this population include the Religious and Spiritual Struggle Scale (Exline et al. 2014) and the Worthiness scale from the Numinous Motivation Inventory (NMI; Piedmont 2017), both of which contain subscales that assess different types of r/s domains of struggle. Recent research has also demonstrated the added utility of open-ended assessment of r/s struggles. Wilt, Takahashi, Jeong, Exline, and Pargament (Wilt et al. 2020) found that open-ended descriptions of religious and spiritual struggles provide valid and reliable data that is associated with, but distinct from, assessments relying on closed-ended items. An open-ended assessment approach may be particularly useful for understanding the r/s struggles of marginalized populations, such as those who have been convicted of a sexual offense.

Given the strong inverse relationship between religious struggle and self-compassion found in this study, previous research that suggests self-compassion may promote desistance (Morley et al. 2016), along with the promise of compassion-based approaches to therapy (Harper et al. 2019; Walton and Hocken 2019), it is important to continue to study the impact of religious struggles on perceptions of the self in this, and other, populations. Research examining the efficacy of acceptance and compassion approaches to therapy for this population should also be examined for their potential impact on religious struggles.

To fully benefit from religion and spirituality, persons who have sexually offended need to be able to rely on inclusion and acceptance within communities of faith. Further, prevention of sexual abuse is, in part, dependent upon widespread accurate information about sexual abuse, community engagement, and well-informed helping professionals (Tabachnick 2013). Research focused on how faith communities might respond to and benefit from programs designed to increase knowledge of the etiology of sexual offending, actual and imagined risk of re-offending, prevention strategies, and approaches for enhancing personal and organizational identities motivated by compassion, may open more church doors for those who have offended while simultaneously enriching the congregational identity along with that of the person who has offended.

**Author Contributions:** Conceptualizations, T.M.R.; Methodology, T.M.R., R.L.P.; Formal Analysis, T.M.R., G.M.M.-R., R.L.P.; Investigation, T.M.R.; Resources, T.M.R.; Data Curation, T.M.R.; Writing—Original Draft Preparation, T.M.R., G.M.M.-R., Writing—Review & Editing, T.M.R., G.M.M.-R., R.L.P.; Funding Acquisition, T.M.R.; Project Administration, T.M.R. All authors have read and agreed to the published version of the manuscript.

**Funding:** Funding: This research was partially funded by the National Board of Certified Counselors Minority Fellowship Program and Loyola University Maryland Kolvenbach Research Grant.

**Conflicts of Interest:** Ralph L. Piedmont receives royalties from the sale of the ASPRES.

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
