# Peer review of "Let Him Who Is without Sin Cast the First Stone: Religious Struggle among Persons Convicted of Sexually Offending"

_religions, doi:10.3390/rel11110546_

Round 1
Reviewer 1 Report
This is an important and worthwhile study which builds on previous research. The article communicates a humane interest in the well-being of men who have caused suffering for others, but who also suffer as a result of their behavior. It is in the interests of society to support these men in order to improve their mentla health and sense of self-worth, as men with "good lives" are less likely to reoffend. THis information may also be of use to ffiath communities to support their efforts to find ways to enable the men to receive support and consolation from religion and spirituality.
The sample size and correlational design have limitations but teh data is valuable and should be shared more widely.
Plesae note the attached observations:
Review Report
Brief Summary
This study explores a series of correlations between religious struggle, personality and mental health in a sample of 30 registered sex offenders. Mean levels of spiritual transcendence were lower than normative samples and greater levels of spiritual struggle were related to higher levels of neuroticism, as well as self-reported shame and other mental health variables such as anxiety and depression. The authors argue that it is in the interest of society, which wants to promote safety, and the well-being of registered sex offenders to lessen their spiritual struggles, as religion is seen as a means of support and re-integration into society for sex offenders. This can also improve their sense of self (less shame) and lessen mental health challenges. These findings have implications for faith groups regarding the development of a more positive response to providing support for registered sex offenders. Mental health professionals who deliver treatment to sex offenders should give consideration to religion and spirituality as part of their treatment approach to sex offenders in order to promote their well-being and support the safety of communities.
Broad Comments:
The article begins with an arresting title which will attract the interest of readers of this journal and may attract readers who would not normally give attention to research regarding sex offenders. The authors have two objectives: to explore the relationship between religious struggles and (a) desistance, and (b) mental health. They state the first objective clearly and concisely in lines 135 – 138:
Despite recent scholarly interest in how r/s might positively or negatively influence desistance, very little is known about how r/s constructs present among those who have sexually offended, or how religious and spiritual struggles might contribute to, or deter, from desistance.
The second objective is stated clearly in lines 579 – 581: “a primary objective of this study was to assess how religion and spirituality were related to mental health among participants.” The authors maintain their focus on these two goals throughout their paper.
Previous research has demonstrated the importance of religion and spirituality in the well-being and recovery of sex offenders. This study takes the research further by exploring the issue of spiritual struggle, and demonstrates significant correlations between personality and mental health. The authors make the case that the experience of spiritual struggle can interfere with the benefits of spiritual support through a personal relationship with the God of their understanding and participation in a faith community which can provide support and a value system that supports desistance from sexual offending. Exclusion from such faith communities can contribute to the mental health challenges that reduce well-being, e.g., depression, anxiety, and hopelessness. These are also shown in greater reported levels of self-shame and lower levels of self-compassion. These findings provide detail and depth to the question of how to promote religious practice and spiritual development for those who have committed sexual offences. The authors make use of appropriate measures and provide data regarding the reliability and merits of their chosen research instruments.
The introduction is clear and makes use of the relevant research literature in this area, demonstrating wide reading in labelling (stigmatization) and desistance studies. The discussion addresses the stated objectives and situates the research in a tradition of humane response to and treatment for sex offenders, and society’s desire to impose appropriate punishment and its need for safety in communities.
There is a concern regarding the issue of statistical power, given the number of measures that are used and the sample size (30). However, the authors address this issue directly and comprehensively in their discussion of the study’s limitations. The correlational design is also limited in terms of statistical strength, but this is what was possible, given the number of participants. The authors address this directly in lines 709 – 712.
They also address the issue of the low alpha reliability found for Religious Crisis with a proposed way to improve this in lines 702 – 709.
However, the data which has been harvested from the study is valuable and important, and provides depth and detail regarding the place of religion and spirituality in the recovery and treatment of sex offenders, and its relationship with mental health outcomes. This data is not available elsewhere, and should be shared with a wider audience The results of this study provide an insight into the difficulties faced by these men and provide support for those who advocate a Good Lives Model (GLM) of treatment to enable these men to re-integrate into society, find ways to participate in a faith community and develop a relationship with the God of their understanding, and, in the process, improve their mental health and feelings of self-worth. The authors demonstrate the key role that religion and spirituality have in such an approach.
Specific comments
Line |
Comment |
20 |
Person: either “a person,” or “persons.” |
42 and 52 |
In line 42 the authors write about leprosy in the present: “people with leprosy have a physiological disease . . .” This continues in line 43: “the social stigma that they have been subjected to . . . the disease is not their fault.” In line 52 the authors change to the past tense, consistent with references in the first paragraph about leprosy in the time of Jesus: “The treatment of persons with leprosy was based on inaccurate assumptions at different times and places in history.” One solution would be to add the following words to line 52: “ . . .just as the ostracism and treatment of persons with leprosy at the time of Jesus was based on inaccurate assumptions . . .” There may be other ways to sort out this small grammatical matter and the authors may see other possibilities. |
46 |
Straightforward – one word or two? |
48 - 49 |
As the second sentence on line 48 stands, it suggests that “Treatment options are limited, environmental and physiological.” This does not seem right to me. Could I suggest the following: “Treatment options are limited, and environmental, physiological and psychological factors are not considered when labeling individuals as “sex offenders.” |
59 |
. . .and the appeal to moral luck |
59 |
The appeal to ignorance can also be a form of denial. |
64 |
In this sentence the word “etiology” is used for hebophilia and pedophilia, and the verb is singular, “remains,” even though it covers both hebophilia and pedophilia, which probably have different etiologies. Might it be more correct to write: “While the precise etiologies of hebophilia and pedophilia remain uncertain, they are believed to result from . . . .?” |
242 |
From attend services: “From attending services?” |
319 Table 2 |
Relationship to victim. “No victim.” I presume that this refers to the “nearly one quarter” of participants who were convicted of viewing sexually explicit images of children.” If this is the case, then I would suggest that another name for the grouping should be found as it is not true that there is “no victim” in these crimes. Possibly, “On-line victim.” See also the category of “Offense,” found later in the table.
The total for relationship to victim is 29. The total for Victim involved in Offence is 30. When we add the various categories in Relationship to Victim (leaving aside “No victim” the total is 22, or 75.9%. The total of “Yes” tallies for “Victim involved in Offense” is 23, or 76.7%. This discrepancy should be corrected (if inaccurate) or explained. Also, it is not correct to say that on-line victims are not involved in the offence. Perhaps they could be labelled as “non- contact victims.”
Victim Years of age: Perhaps include a category of “Unknown” with N = 7. It might be mentioned that since these men were found guilty of a criminal activity that the age of these victims must have been under 18 years of age. |
327 |
Participants requires a possessive inverted comma: participants’ |
438 |
openness |
449 |
As far as I can see “STS” has not been introduced as an acronym prior to this point. This acronym should be explained, and the full name given so that the reference is clear. Apologies if I have overlooked an earlier reference. |
464 |
Wartting et al (2013) does not appear in the bibliography. |
532 – 533 and 538. |
Singular and plural “individuals . . . face in their lives . . . |
543 |
ASPIRES |
565 |
Spelling of “associateid.” |
666 |
Central tenant – should be “Central tenet?” |
682 |
Persons: should be “person’s.” |
699 |
“Geary, Ciarrocchi and Scheers, 2004” is the same reference as “Geary et al. 2004.” It will be simpler to write: “Geary, Ciarrocchi and Scheers, 2004, 2006.,” or “Geary et al 2004, 2006.” As in line566 |
692 |
“intimated.” I was not familiar with this use of “intimated,” and looked it up in Dictionary.com, where I learned that it can mean, “associated in close personal relations,” or “characterized by or involving warm friendship or a personally close or familiar association or feeling, or “very private: closely personal.” This word, in some of its possible meanings, appears to offer a reasonable hypothesis regarding the motivation of (some of) those who participated. It could also be true that those who participated were more intimated, and that the self-selection based on a mailed invitation did not appeal to them. The word “intimated” appears to relate to those who are (a) very private and (b) characterized by warm friendship. The first definition could lead to non- participation, and the second definition could lead to a desire to make connection and complete the survey. I suggest that this word be clarified, and more reflection given to the hypotheses regarding motivation to participate. |
698 |
Indicitive: should this be “Indicative?” |
698 |
Importantce: remove the second “t.” |
719 |
Religions: Should this be “religious?” |
Author Response
This is an important and worthwhile study which builds on previous research. The article communicates a humane interest in the well-being of men who have caused suffering for others, but who also suffer as a result of their behavior. It is in the interests of society to support these men in order to improve their mental health and sense of self-worth, as men with "good lives" are less likely to reoffend. This information may also be of use to faith communities to support their efforts to find ways to enable the men to receive support and consolation from religion and spirituality.
The sample size and correlational design have limitations but the data is valuable and should be shared more widely.
Response: Thank you for noting that despite the limitations of the sample size and correlational design, the article is written from a humane perspective and acknowledges the suffering of those who have caused suffering as well as the important role of supporting these men to help them move toward connected and purposeful lives not only for their sakes but for the overall good of society in terms of reducing the likelihood of reoffending. We also appreciate your noting the possible benefit to faith communities which may be looking for guidance about how to support these men within the context of their faith.
The article begins with an arresting title which will attract the interest of readers of this journal and may attract readers who would not normally give attention to research regarding sex offenders. The authors have two objectives: to explore the relationship between religious struggles and (a) desistance, and (b) mental health. They state the first objective clearly and concisely in lines 135 – 138:
Despite recent scholarly interest in how r/s might positively or negatively influence desistance, very little is known about how r/s constructs present among those who have sexually offended, or how religious and spiritual struggles might contribute to, or deter, from desistance.
The second objective is stated clearly in lines 579 – 581: “a primary objective of this study was to assess how religion and spirituality were related to mental health among participants.” The authors maintain their focus on these two goals throughout their paper.
Response: We appreciate the feedback about the title and sincerely hope it does indeed peak interest in some who may not have otherwise been interested in learning more about persons who have sexually offended. Thank you for noting the clarity of the paper objectives.
Previous research has demonstrated the importance of religion and spirituality in the well-being and recovery of sex offenders. This study takes the research further by exploring the issue of spiritual struggle, and demonstrates significant correlations between personality and mental health. The authors make the case that the experience of spiritual struggle can interfere with the benefits of spiritual support through a personal relationship with the God of their understanding and participation in a faith community which can provide support and a value system that supports desistance from sexual offending. Exclusion from such faith communities can contribute to the mental health challenges that reduce well-being, e.g., depression, anxiety, and hopelessness. These are also shown in greater reported levels of self-shame and lower levels of self-compassion. These findings provide detail and depth to the question of how to promote religious practice and spiritual development for those who have committed sexual offences. The authors make use of appropriate measures and provide data regarding the reliability and merits of their chosen research instruments.
Response: We agree with findings from previous research that religion and spirituality can promote recovery among of and desistence from sexual offending and appreciate your highlighting this and noting that this study offers additional data as religious struggle among this population has not received attention to date.
The introduction is clear and makes use of the relevant research literature in this area, demonstrating wide reading in labeling (stigmatization) and desistance studies. The discussion addresses the stated objectives and situates the research in a tradition of humane response to and treatment for sex offenders, and society’s desire to impose appropriate punishment and its need for safety in communities.
Response: Thank you for noting the relevance and extent of the literature review and for reiterating the humane approach to those who have offended within the context of public safety and appropriate punishments. Based on the reviewer’s comments that the introduction was clear, uses relevant literature, addresses objectives, and situates the research within the context of humane treatment of those who have offended, appropriate punishment, and community safety we did not make substantive changes to the introduction. All specific comments in the introduction were addressed and are outlined below.
There is a concern regarding the issue of statistical power, given the number of measures that are used and the sample size (30). However, the authors address this issue directly and comprehensively in their discussion of the study’s limitations. The correlational design is also limited in terms of statistical strength, but this is what was possible, given the number of participants. The authors address this directly in lines 709 – 712.
Response: Yes, the authors agree that sample size and resulting statistical power are of concern. We appreciate that the reviewer stated this same concern and also that the reviewer’s assessment is that these concerns were addressed in the limitations section. The discussion of the sample size in the limitations section have been revised and are outlined below under “Specific comments.”
They also address the issue of the low alpha reliability found for Religious Crisis with a proposed way to improve this in lines 702 – 709.
Response: Again, the authors concur that the alpha reliability for Religious Crisis was lower than desired.
However, the data which has been harvested from the study is valuable and important, and provides depth and detail regarding the place of religion and spirituality in the recovery and treatment of sex offenders, and its relationship with mental health outcomes. This data is not available elsewhere, and should be shared with a wider audience The results of this study provide an insight into the difficulties faced by these men and provide support for those who advocate a Good Lives Model (GLM) of treatment to enable these men to re-integrate into society, find ways to participate in a faith community and develop a relationship with the God of their understanding, and, in the process, improve their mental health and feelings of self-worth. The authors demonstrate the key role that religion and spirituality have in such an approach.
Response: We thank the reviewer for the complimentary remarks and appreciate the perspective that despite the limitations, the data contribute to important and understudied areas related to the impact of religion and spirituality in the recovery, treatment, and mental health of persons who have sexually offended. We are also grateful that the reviewer noted the data may be helpful for advocates of the GLM.
Specific Comments
Line |
Comment |
20 |
Person: either “a person,” or “persons.” Line 17 (previous line 20): person changed to “persons. |
42 and 52 |
In line 42 the authors write about leprosy in the present: “people with leprosy have a physiological disease . . .” This continues in line 43: “the social stigma that they have been subjected to . . . the disease is not their fault.” In line 52 the authors change to the past tense, consistent with references in the first paragraph about leprosy in the time of Jesus: “The treatment of persons with leprosy was based on inaccurate assumptions at different times and places in history.” One solution would be to add the following words to line 52: “ . . .just as the ostracism and treatment of persons with leprosy at the time of Jesus was based on inaccurate assumptions . . .” There may be other ways to sort out this small grammatical matter and the authors may see other possibilities. Lines 38-39 (previous line 42-43) and line 51 (previously line 52) : The text has been changed to correct the grammatical inconsistencies to reflect past tense throughout. Thank you for pointing this out. |
46 |
Straightforward – one word or two? Line 46: Straight forward changed to one word: straightforward. |
48 - 49 |
As the second sentence on line 48 stands, it suggests that “Treatment options are limited, environmental and physiological.” This does not seem right to me. Could I suggest the following: “Treatment options are limited, and environmental, physiological and psychological factors are not considered when labeling individuals as “sex offenders.” Line 47 (previously lines 48-49): No, it was not correct. Thank you for identifying it and for your recommendation. The text has been revised. |
59 |
. . .and the appeal to moral luck. Line 56 (previously line 59): “the” added. |
59 |
The appeal to ignorance can also be a form of denial. Lines 58 – 61 (previously line 56): Thank you for pointing this out. Text has been added to note that the appeal to ignorance can be a form of denial and that it may also be a consideration in the early phases of the desistence process as a means to negotiate stigma.
|
64 |
In this sentence the word “etiology” is used for hebophilia and pedophilia, and the verb is singular, “remains,” even though it covers both hebophilia and pedophilia, which probably have different etiologies. Might it be more correct to write: “While the precise etiologies of hebophilia and pedophilia remain uncertain, they are believed to result from . . . .?” Line 65 (previously line 64): Again, thank you. The text has been revised per the reviewer’s suggestion. |
242 |
From attend services: “From attending services?” Line 245 (previously line 242): Changed from “attend” to “attending.” |
319 Table 2 |
Relationship to victim. “No victim.” I presume that this refers to the “nearly one quarter” of participants who were convicted of viewing sexually explicit images of children.” If this is the case, then I would suggest that another name for the grouping should be found as it is not true that there is “no victim” in these crimes. Possibly, “On-line victim.” See also the category of “Offense,” found later in the table. Lines 323 – 326 (Table 2 – Previously beginning on line 319): Yes, it does refer to those who were convicted of viewing sexually explicit images of children. Your point is well taken and greatly appreciated. Revisions have been made to the table to more accurately reflect the victimization of the children whose images are used. A note has been added to Table 2 to clarify “unknown” offense. The total for relationship to victim is 29. The total for Victim involved in Offence is 30. When we add the various categories in Relationship to Victim (leaving aside “No victim” the total is 22, or 75.9%. The total of “Yes” tallies for “Victim involved in Offense” is 23, or 76.7%. This discrepancy should be corrected (if inaccurate) or explained. Also, it is not correct to say that on-line victims are not involved in the offence. Perhaps they could be labelled as “non- contact victims.” Lines 323 – 326 (Table 2 – Previously beginning on line 319):There was 1 missing response to question asking for relationship to victim. The title of column has been changed to indicate the percentages are valid percents. “Non-contact victim” has been added to the table. The authors are quite appreciative of these important corrections to clarify the nature of victimization in sexually explicit images of children. Thank you! Victim Years of age: Perhaps include a category of “Unknown” with N = 7. It might be mentioned that since these men were found guilty of a criminal activity that the age of these victims must have been under 18 years of age. Lines 325 – 326 (Table 2 -Previously beginning on line 319): A note has been added to Table 2 to clarify “unknown” age. |
327 |
Participants requires a possessive inverted comma: participants’ Line 332 (Previously line 327): possessive inverted comma added. |
438 |
openness Line 462 (Previously line 438): “s” added. |
449 |
As far as I can see “STS” has not been introduced as an acronym prior to this point. This acronym should be explained, and the full name given so that the reference is clear. Apologies if I have overlooked an earlier reference. Line 477 (Previously line 449): This was indeed the first time STS was in the paper. In the discussion of the ASPIRES in section 2.3.1. it was referred to as ST. The reference the spiritual transcendence scale has been changed to reflect previous references to the scale – ST. |
464 |
Wartting et al (2013) does not appear in the bibliography. Line 1012: The Wartting reference was attached to the previous reference. Hard return was added to separate it. |
532 – 533 and 538. |
Singular and plural “individuals . . . face in their lives . . . Lines 562 – 567 (Previously lines 532 – 538): Singular changed to plural. |
543 |
ASPIRES Line 574 (Previously line 543): “S” added. |
565 |
Spelling of “associateid.” Line 594 (Previously line 565): Spelling corrected. |
666 |
Central tenant – should be “Central tenet?” Line 691 (Previously line 666): Spelling corrected. |
682 |
Persons: should be “person’s.” Line 706 (Previously line 682): “person’s” added. |
699 |
“Geary, Ciarrocchi and Scheers, 2004” is the same reference as “Geary et al. 2004.” It will be simpler to write: “Geary, Ciarrocchi and Scheers, 2004, 2006.,” or “Geary et al 2004, 2006.” As in line566 Line 727 (Previously line 699): Citation changed. |
692 |
“intimated.” I was not familiar with this use of “intimated,” and looked it up in Dictionary.com, where I learned that it can mean, “associated in close personal relations,” or “characterized by or involving warm friendship or a personally close or familiar association or feeling, or “very private: closely personal.” This word, in some of its possible meanings, appears to offer a reasonable hypothesis regarding the motivation of (some of) those who participated. It could also be true that those who participated were more intimated, and that the self-selection based on a mailed invitation did not appeal to them. The word “intimated” appears to relate to those who are (a) very private and (b) characterized by warm friendship. The first definition could lead to non- participation, and the second definition could lead to a desire to make connection and complete the survey. I suggest that this word be clarified, and more reflection given to the hypotheses regarding motivation to participate. Lines 717 – 721 (Previously line 692): “intimated” was a typo. It was intended to be “intimidated.” The word intimidated has replaced intimated and the discussion has been revised to reflect the original intent. Our apologies for the confusion, and appreciation for your thoughtful comments. |
698 |
Indicitive: should this be “Indicative?” Line 727 (Previously line 698): Spelling corrected. |
698 |
Importantce: remove the second “t.” Line 727 (Previously line 698): “t” removed. |
719 |
Religions: Should this be “religious?” Line 751 (Previously line 719): Changed to “religious” |
Reviewer 2 Report
I do not have comments for revisions.
Author Response
Thank you for your enthusiasm about the topic of the paper and your careful evaluation of our original submission.